# Revisiting Quantum Algorithms for Linear Regressions: Quadratic Speedups without Data-Dependent Parameters

## Abstract

Linear regression is one of the most fundamental linear algebra problems. Given a dense matrix $A \in \mathbb{R}^{n \times d}$ and a vector $b$, the goal is to find $x'$ such that $\|Ax' - b\|_2^2 \leq (1 + \epsilon) \min_x \|Ax - b\|_2^2$. The best classical algorithm takes $O(nd) + \mathrm{poly}(d/\epsilon)$ time [Clarkson and Woodruff STOC 2013, Nelson and Nguyen FOCS 2013]. On the other hand, quantum linear regression algorithms can achieve exponential quantum speedups, as shown in [Wang *Phys. Rev. A 96*, 012335, Kerenidis and Prakash ITCS 2017, Chakraborty, Gilyén and Jeffery ICALP 2019]. However, the running times of these algorithms depend on some quantum linear algebra-related parameters, such as $\kappa(A)$, the condition number of $A$. In this work, we develop a quantum algorithm that runs in $\widetilde{O}(\epsilon^{-1}\sqrt{n}d^{1.5}) + \mathrm{poly}(d/\epsilon)$ time and outputs a classical solution. It provides a quadratic quantum speedup in $n$ over the classical lower bound without any dependence on data-dependent parameters. In addition, we also show our result can be generalized to multiple regression and ridge linear regression.

## 1 Introduction

Linear regression is one of the fundamental problems in machine learning and data science (Hilary, 1967; Yan & Su, 2009; Freedman, 2009). It is a statistical method that models the relationship between a dependent variable and one or more independent variables. Multiple regression is an extension of linear regression that predicts a target variable using multiple feature variables. They have many applications across different areas such as predictive analysis (Khine & Nyunt, 2019; Vovk et al., 2009), economics (Nizam et al., 2020; Porta et al., 2008; Acemoglu et al., 2001), marketing (Berger & Nasr, 1998), finance (Götze et al., 2023), healthcare (Lukong & Jafaru, 2021; Kan et al., 2019; Valsamis et al., 2019; Tomar & Agarwal, 2013), education (Reddy & Sarma, 2015; Baker & Richards, 1999; Olsen et al., 2020), social sciences (Uyanik & Güler, 2013; Yin, 2023), sports analytics (Sarlis & Tjortjis, 2020; Chu & Wang, 2019), manufacturing (Chiarini & Brunetti, 2019; Baturynska & Martinsen, 2021), and quality control (QIU & Bo, 2012). The definition of linear regression is as follows:

**Definition 1.1** (Linear Regression). *Given a matrix $A \in \mathbb{R}^{n \times d}$ and a vector $b \in \mathbb{R}^n$, we let $\epsilon \in (0, 1)$ denote an accuracy parameter. The goal is to output a vector $x \in \mathbb{R}^d$ such that $\|Ax - b\|_2^2 \leq (1 + \epsilon) \min_{x' \in \mathbb{R}^d} \|Ax' - b\|_2^2$.*

The state-of-the-art algorithms for solving linear regression are due to Clarkson & Woodruff (2013); Nelson & Nguyên (2013), where the running time is $O(nd) + \mathrm{poly}(d/\epsilon)$. The formal definition of multiple regression is as follows:

**Definition 1.2** (Multiple Regression). *Given two matrices $A \in \mathbb{R}^{n \times d}$ and $B \in \mathbb{R}^{n \times N}$, we let $\epsilon \in (0, 1)$ denote an accuracy parameter. The goal is to output a matrix $X \in \mathbb{R}^{d \times N}$ such that $\|AX - B\|_F \leq (1 + \epsilon) \min_{X' \in \mathbb{R}^{d \times N}} \|AX' - B\|_F$.*

Ridge regression is a regularized version of linear regression that adds an $\ell_2$ penalty to the regression coefficients, preventing overfitting. This property makes it well-suited for handling high-dimensional data, where feature collinearity is common. In machine learning, ridge regression serves as a common

baseline and benchmark method. Several studies have analyzed ridge regression concerning high-dimensional data and models (Dobriban & Wager, 2018; Maronna, 2011), feature selection (Paul & Drineas, 2016; Zhang et al., 2018; Cawley, 2008), and regularization path algorithms (Friedman et al., 2010). Moreover, it has found extensive use in diverse applications such as image recognition (An et al., 2007; Xue et al., 2009), natural language processing (Bedo et al., 2006; Liu, 2021), and bioinformatics (Xu et al., 2020; Cule et al., 2011; Bedo et al., 2006).

**Definition 1.3** (Ridge Regression). *Given a matrix $A \in \mathbb{R}^{n \times d}$ and a vector $b \in \mathbb{R}^n$, we let $\epsilon \in (0, 1)$ denote an accuracy parameter and let $\lambda > 0$ denote a regularization parameter. The goal is to output a vector $x \in \mathbb{R}^d$ such that $\|Ax - b\|_2^2 + \lambda\|x\|_2^2 \leq (1 + \epsilon)\min_{x' \in \mathbb{R}^d}(\|Ax' - b\|_2^2 + \lambda\|x'\|_2^2)$.*

In this paper, we study the quantum algorithms for the linear regression problem and its variations, including ridge regression and multiple regression. Quantum computing is a rapidly advancing technology, and we now have quantum computers with dramatically increasing capabilities that are on the cusp of achieving quantum advantages over classical computers. It is thus a pertinent question whether quantum computers can accelerate solving classical machine learning optimizations like linear regression. Quantum algorithms for linear regression have been studied for a long time (Wiebe et al., 2012; Schuld et al., 2016; Wang, 2017; Kerenidis & Prakash, 2017; Chakraborty et al., 2019; Shao & Xiang, 2020; Chen & de Wolf, 2021; Shao, 2023; Chen et al., 2023; Chakraborty et al., 2023). However, the majority of existing algorithms rely on quantum linear algebra techniques, which harbor noteworthy limitations. Specifically, their time complexities hinge on the condition number $\kappa$ of the input matrix. This predicament impedes a direct comparison with state-of-the-art classical methods, whose runtimes remain independent of $\kappa$. Consequently, these quantum algorithms can only guarantee acceleration over classical ones for instances featuring well-conditioned input matrices.

Overcoming this conditional dependence is an open question. We would like quantum regression algorithms that can provably achieve speedups for any input matrix, not just "easy" ones. Developing such algorithms requires departing from the quantum linear algebra framework and exploring novel techniques. In our work, we make progress in this direction by proposing quantum algorithms for linear regression, ridge regression, and multiple regression based on leverage score distribution. Our approach achieves a runtime proportional to the square root of the data dimension $n$, without the dependence on the condition number $\kappa$. This marks the first unconditional acceleration for these three regression problems in comparison to the best-known classical algorithm. In the classical setting, it is well known that solving linear regression requires $\Omega(n)$ time (Clarkson & Woodruff, 2013; Nelson & Nguyên, 2013). In the quantum setting, it has been known for a while that $\Omega(\sqrt{n} + d)$ time is a lower bound (Wang, 2017; Shao, 2023). Thus, we can ask the following question:

*Is it possible to solve linear regression in $O_d(\sqrt{n})$ time and without paying matrix-dependent parameters (e.g., $\kappa(A)$)?*

## 1.1 OUR RESULTS

We provide a positive answer to this question. The main contribution of our work is to propose a quantum algorithm that solves linear regression in $O_d(\sqrt{n})$ time while the classical algorithm requires $\Omega_d(n)$ time. Notice that the complexity of our algorithm does not have any data-dependent parameter. For comparison, the quantum linear regression algorithms proposed by Wang (2017); Kerenidis & Prakash (2017); Chakraborty et al. (2019) have a time complexity $\text{poly}(\log n, d, \kappa(A), \epsilon^{-1})$. On the other hand, there exists a series of works on developing "quantum-inspired" algorithms for linear regression problems, which show that classical algorithms can also achieve $\log(n)$-dependence by assuming some sampling access to the input matrix. However, the time complexities of these algorithms have large polynomial dependence on some matrix parameters. In particular, Chia et al. (2022) presented a quantum-inspired algorithm that runs in $O((\|A\|_F/\|A\|)^6 \kappa(A)^{28})$ time. Further, Gilyén et al. (2020) improved to $O((\|A\|_F/\|A\|)^6 \kappa(A)^{12})$ time. We show our results for linear regression and its generalizations (multiple regression and ridge regression) in more detail below.

**Linear regression.** We develop a quantum algorithm solving linear regression with classical output.

**Theorem 1.4** (Quantum algorithm for linear regression). *Let $\epsilon \in (0, 1)$. Let $\omega \approx 2.37$ denote the exponent of matrix multiplication. Given a matrix $A \in \mathbb{R}^{n \times d}$ and $b \in \mathbb{R}^n$, there is a quantum algorithm that outputs $x \in \mathbb{R}^d$ such that $\|Ax - b\|_2 \leq (1 + \epsilon)\min_{x' \in \mathbb{R}^d} \|Ax' - b\|_2$, which takes*

$\widetilde{O}(\sqrt{n}d/\epsilon)$ *row queries to $A$ and $\widetilde{O}(\sqrt{n}d^{1.5}/\epsilon + d^{\omega}/\epsilon)$ time, where $r$ is the row of sparsity of matrix $A$ and $r \leq d$. The success probability is 0.999.*

**Multiple regression.**   We also improve the classical multiple linear regression algorithm from $O(nd) + N\operatorname{poly}(d)$ (Clarkson & Woodruff, 2013; Nelson & Nguyên, 2013) to $\widetilde{O}(\sqrt{n}d^{1.5}) + N\operatorname{poly}(d)$, where $N$ is the number of columns of the matrix $B$ (see Definition 1.2)

**Theorem 1.5** (Quantum algorithm for multiple regression)**.** *Let $\epsilon \in (0, 1)$. Let $\omega \approx 2.37$ denote the exponent of matrix multiplication. Given a matrix $A \in \mathbb{R}^{n \times d}$ with row sparsity $r$, where $r \leq d$, $B \in \mathbb{R}^{n \times N}$, there is a quantum algorithm that outputs $X \in \mathbb{R}^{d \times N}$ such that $\|AX - B\|_F \leq (1 + \epsilon)\min_{X' \in \mathbb{R}^{d \times N}} \|AX' - B\|_F$, which takes $\widetilde{O}(\sqrt{n}d/\epsilon)$ row queries to $A$ and $\widetilde{O}(\sqrt{n}d^{1.5}/\epsilon + d^{\omega}/\epsilon + Nd^{\omega-1}/\epsilon)$ time. The success probability is 0.999.*

**Ridge regression.**   For ridge regression, the previous best ridge regression algorithm is due to Avron et al. (2017), which has a running time $O(nd) + \operatorname{poly}(d, \mathsf{sd}_\lambda(A), 1/\epsilon)$, where $\mathsf{sd}_\lambda(A)$ is the statistical dimension of $A$. In quantum, Shao (2023) gave a quantum algorithm with classical outputs that has a linear dependence in $n$ in the worst case. Shao & Xiang (2020); Chen et al. (2023); Chakraborty et al. (2023) showed quantum algorithms that can prepare quantum states encoding the solution. In particular, the algorithm by Chakraborty et al. (2023) has a cost $\widetilde{O}(\kappa + \frac{\alpha_A}{\sqrt{\lambda}})$, where $\kappa = 1 + \frac{\|A\|}{\lambda}$ and $\alpha_A \leq \|A\|_F$ is a data-dependent parameter. We note that these algorithms (with quantum outputs) are incomparable to ours since the output formats are different. Chen & de Wolf (2021) studied quantum algorithms for LASSO (linear regression with $\ell_1$-constraint) and ridge regressions. However, they focused on improving the $d$-dependence and only considered the regime when $n = O(\log(d)/\epsilon^2)$. We present a quantum algorithm that runs in $\widetilde{O}(\sqrt{n \cdot \mathsf{sd}_\lambda(A)}d) + \operatorname{poly}(\mathsf{sd}_\lambda(A), 1/\epsilon)$ time.

**Theorem 1.6** (Quantum algorithm for ridge regression)**.** *Given a matrix $A \in \mathbb{R}^{n \times d}$ and $b \in \mathbb{R}^n$, we let $\mathsf{sd}_\lambda(A)$ denote the statistical dimension of matrix $A$ (see Definition 3.8), $\epsilon \in (0, 1)$, and $\lambda > 0$ denote a regularization parameter. There is a quantum algorithm that outputs $x \in \mathbb{R}^d$ such that $\|Ax - b\|_2^2 + \lambda\|x\|_2^2 \leq (1 + \epsilon)\min_{x' \in \mathbb{R}^d}(\|Ax' - b\|_2^2 + \lambda\|x'\|_2^2)$, which takes $\widetilde{O}(\sqrt{n \cdot \mathsf{sd}_\lambda(A)}/\epsilon)$ row queries to $A$ and $\widetilde{O}(\sqrt{n \cdot \mathsf{sd}_\lambda(A)}d/\epsilon + \operatorname{poly}(d, \mathsf{sd}_\lambda(A), 1/\epsilon))$ time, with 0.999 success probability.*

**Roadmap.**   In Section 2, we introduce the related work. In Section 3, we present the preliminary of our work. In Section 4, we analyze the linear regression problem and the multiple regression problem and present the formal version of our main results. In Section 5, we analyze the ridge regression problem and present the formal version of our main result. In Section 6, we make a conclusion for this paper and discuss the limitations and societal impacts.

## 2   RELATED WORK

**Quantum optimization algorithms**   Optimization is one of the promising areas to demonstrate quantum advantages. Since the groundbreaking result by Harrow et al. (2008) on the quantum linear system solver, a significant number of works (such as Childs et al. (2017); Low & Chuang (2019); Gilyén et al. (2019); Chakraborty et al. (2019)) have focused on developing quantum algorithms to accelerate linear algebra operations. These algorithms are commonly referred to as "quantum linear algebra". Unlike classical numerical linear algebra algorithms whose solutions are classical vectors or matrices, the outputs of quantum linear algebra algorithms are usually quantum states that encode the solution. Specifically, it is possible to represent an $n$-dimensional vector as a $O(\log(n))$-qubit quantum state. This allows a quantum computer to solve problems exponentially faster than classical computers. Based on the quantum linear algebra approach, several quantum optimization algorithms have been developed. In addition to the quantum linear regression algorithms mentioned before, there has been a long list of work on fast quantum linear programming (LP) and semi-definite programming (SDP) solvers (Brandao & Svore, 2017; Apeldoorn et al., 2017; Brandão et al., 2019; Apeldoorn & Gilyén, 2019; Kerenidis & Prakash, 2020; Huang et al., 2022). On the other hand, based on Jordan's algorithm (Jordan, 2005) for computing gradients in quantum, van Apeldoorn et al. (2020); Chakrabarti et al. (2020) showed quantum speedups of optimizing a convex function over an $n$-dimensional convex body. Other quantum optimization algorithms include finding the Nash equilibrium of a zero-sum game (Bouland et al., 2023; Vasconcelos et al., 2023), sub-modular optimization (Hamoudi et al., 2019), approximate convex optimization (Li & Zhang,

2022), stochastic optimization (Sidford & Zhang, 2023), escaping from saddle points (Zhang et al., 2021a). Another approach for quantum optimization is via variational quantum algorithms such as the variational quantum eigensolver (VQE) (Peruzzo et al., 2014) or the quantum approximate optimization algorithm (QAOA) (Farhi et al., 2014). A large number of algorithms have been developed to solve combinatorial optimization problems, e.g., Guerreschi & Matsuura (2019); Basso et al. (2021); Zhang et al. (2021b). The approach requires only a small amount of quantum resources and can be implemented in a real-world device soon. However, a rigorous analysis on the performance and quantum advantage for this approach remains open.

**Quantum machine learning** Quantum machine learning (QML) is a field that examines how quantum computing can enhance machine learning. Several quantum algorithms have been proposed to provide speedups or improved capabilities compared to classical machine learning approaches, such as clustering (Harrow, 2020; Doriguello et al., 2023), boosting (Arunachalam & Maity, 2020), support vector machine (Rebentrost et al., 2014), principal component analysis (Lloyd et al., 2014), statistical query learning (Arunachalam et al., 2020), etc. However, the seminal work by Tang (2018) showed that some quantum linear algebra-based QML algorithms (such as the quantum recommendation system (Kerenidis & Prakash, 2016)) can be "de-quantized" by some classically samplable data structures. Later, more quantum-inspired algorithms have been proposed for principal component analysis (Tang, 2018), low-rank approximation (Gilyén et al., 2018; Chia et al., 2020), etc. Another approach of QML is to use parameterized quantum circuits and hybrid quantum-classical training strategies to learn from classical or quantum data, such as quantum neural networks (Farhi & Neven, 2018; Cong et al., 2019; Beer et al., 2020; Abbas et al., 2021), quantum kernel methods (Mengoni & Di Pierro, 2019; Schuld & Killoran, 2019; Bartkiewicz et al., 2020).

**Classical Linear Algebra** Given such a family $\Pi$, it is natural to apply an $S$ to $A$ and then solve the smaller problem directly. This is the so-called *sketch-and-solve* paradigm. Sketch-and-solve has led to the development of fast algorithms for many problems, such as linear regression (Clarkson & Woodruff, 2013; Nelson & Nguyên, 2013), low rank approximation with Frobenious norm (Clarkson & Woodruff, 2013; Nelson & Nguyên, 2013), fairness of regression (Song et al., 2023a), matrix CUR decomposition (Boutsidis & Woodruff, 2014; Song et al., 2017; 2019c), weighted low rank approximation (Razenshteyn et al., 2016; Song et al., 2023d), entrywise $\ell_1$ norm low rank approximation (Song et al., 2017; 2019b), tensor regression (Song et al., 2021a; Reddy et al., 2022; Diao et al., 2018; 2019; Deng et al., 2023b), tensor low rank approximation (Song et al., 2019c), general norm column subset selection (Song et al., 2019a), low rank matrix completion (Gu et al., 2023), designing an efficient neural network training method (Qin et al., 2023b), and attention regression problem (Song et al., 2023f; Gao et al., 2023a).

## 3 PRELIMINARIES

In Section 3.1, we introduce the definitions and properties related to the subspace embedding and approximate matrix product. In Section 3.2, we formally define leverage score distribution. In Section 3.3, we formally define the statistical dimension. In Section 3.4, we present a quantum tool for subspace embedding.

**Notation.** We define $[n] := \{1, 2, 3, \ldots, n\}$ and the $\ell_2$ norm of $x$, $\|x\|_2 := \sqrt{\sum_{i=1}^n x_i^2}$. $A_{i,*} \in \mathbb{R}^d$ is the $i$-th row of $A$, and $A_{*,j} \in \mathbb{R}^n$ is the $j$-th column of $A$. Given $y \in \mathbb{R}^d$ with $\|y\|_2 = 1$, we define the spectral norm of $A$, $\|A\| := \max_{y \in \mathbb{R}^d} \|Ay\|_2$. The Frobenius norm of $A$ is $\|A\|_F := \sqrt{\sum_{i=1}^n \sum_{j=i}^d |A_{i,j}|^2}$. The $\ell_0$ norm of $A$, $\|A\|_0 \in \mathbb{R}$ is the number of nonzero entries in $A$. $I_d$ is the $d \times d$ identity matrix. $A^\top \in \mathbb{R}^{d \times n}$ denotes the transpose of the matrix $A$, and $A^\dagger$ denotes the pseudoinverse of $A$. Given two symmetric matrices $B, C \in \mathbb{R}^{n \times n}$, we use $B \preceq C$ to represent that the matrix $C - B$ is positive semidefinite (or PSD), namely for all $x \in \mathbb{R}^n$, we have $x^\top (C - B)x \geq 0$.

### 3.1 DEFINITIONS OF SE AND AMP

In this section, we introduce key concepts that will be central to proving the guarantees of our quantum algorithms for regression. Specifically, we formally define two main concepts–subspace embedding

and approximate matrix product. Subspace embedding is when multiplying by a sketching matrix approximately preserves the geometry or "norms" of vectors from a given subspace. Approximate matrix product means that multiplying a sketch by matrices $A$ and $B$ roughly preserves the Frobenius or spectral norm as if $A$ was directly multiplied by $B$.

**Definition 3.1** (Subspace embedding, (Sarlos, 2006))**.** *Let $\epsilon, \delta \in (0, 1)$. Let $n > d$. Given a matrix $U \in \mathbb{R}^{n \times d}$ which is an orthonormal basis (i.e., $U^\top U = I_d$), we say $S \in \mathbb{R}^{m \times n}$ is an $\mathsf{SE}(\epsilon, \delta, n, d)$ subspace embedding for $U$ if $\|SUx\|_2^2 = (1 \pm \epsilon)\|Ux\|_2^2$, holds with probability $1 - \delta$, which is equivalent to $\|U^\top S^\top SU - U^\top U\| \leq \epsilon$.*

In general, if $S$ does not depend on $U$, then we call it oblivious subspace embedding. In most places of this paper, our $S$ does depend on $U$. Therefore, we do not use "oblivious" in the definition like other papers (Song et al., 2023e).

**Definition 3.2** (Frobenius norm approximate matrix product, (Woodruff, 2014))**.** *Let $\epsilon, \delta \in (0, 1)$. We say $S \in \mathbb{R}^{m \times n}$ is $\mathsf{FAMP}(\epsilon, \delta, n, d)$ Approximate Matrix Product for $A \in \mathbb{R}^{n \times d}$ if for any $B \in \mathbb{R}^{n \times N}$ we have $\|A^\top S^\top SB - A^\top B\|_F^2 \leq \epsilon^2 \cdot \|A\|_F^2 \cdot \|B\|_F^2$ holds with probability $1 - \delta$.*

Here matrix $B$ has to have the same number of rows as $A$. However, $B$ does not necessarily have the same number of columns as $A$.

**Definition 3.3** (Spectral norm approximate matrix product, see Theorem 17 in Avron et al. (2017))**.** *Let $\epsilon, \delta \in (0, 1)$. We say $S \in \mathbb{R}^{m \times n}$ is $\mathsf{SAMP}(\epsilon, \delta, n, d)$ Approximate Matrix Product for $A \in \mathbb{R}^{n \times d}$ if for any $B \in \mathbb{R}^{n \times N}$ we have $\|A^\top S^\top SB - A^\top B\| \leq \epsilon \cdot \|A\| \cdot \|B\|$ holds with probability $1 - \delta$.*

Here matrix $B$ has to have the same number of rows as $A$. However, $B$ is not necessarily to have the same number of columns as $A$. Due to the page limit, we delay the proof of Claim 3.4 to Section B.1.

**Claim 3.4.** *Let $A \in \mathbb{R}^{n \times d}$, $U$ denote the orthonormal basis of $A$, and $D$ denote a diagonal matrix such that $\|DAx\|_2^2 = (1 \pm \epsilon)\|Ax\|_2^2$ for all $x$. Then, we have*

$$\|DUx\|_2^2 = (1 \pm \epsilon)\|Ux\|_2^2.$$

### 3.2 Leverage Score Distribution

We introduce leverage score (see Definition 3.5) and leverage score distribution (see Definition 3.7), which are well-known concepts in numerical linear algebra. We provide definitions that quantify the leverage score of a matrix row as its squared Euclidean norm under an orthonormal transformation of the matrix. Additionally, we define a leverage score distribution as a probability distribution that samples rows with probabilities proportional to these row leverage scores. Intuitively, leverage scores control how much influence each row vector has in spanning the column space.

**Definition 3.5** (Leverage score, see Definition B.28 in Song et al. (2019c) as an example)**.** *Given a matrix $A \in \mathbb{R}^{n \times d}$, we let $U \in \mathbb{R}^{n \times d}$ denote the orthonormal basis of $A$. We define $\sigma_i := \|U_{i,*}\|_2^2$ for each $i \in [n]$. We say $\sigma \in \mathbb{R}^n$ is the leverage score of $A \in \mathbb{R}^{n \times d}$.*

**Fact 3.6.** *It is well known that $\sum_{i=1}^n \sigma_i = d$.*

**Definition 3.7** ($D \sim \mathsf{LS}(A)$, see Definition B.29 in Song et al. (2019c) as an example)**.** *Let $c > 1$ denote some universal constant. For each $i \in [n]$, we define $p_i := c \cdot \sigma_i/d$. Let $q \in \mathbb{R}^n$ be the vector that $q_i \geq p_i$. Let $m$ denote the sparsity of diagonal matrix $D \in \mathbb{R}^{n \times n}$. We say a diagonal matrix $D$ is a sampling and rescaling matrix according to leverage score of $A$ if for each $i \in [n]$, $D_{i,i} = \frac{1}{\sqrt{mq_i}}$ with probability $q_i$ and $0$ otherwise. (Note that each $i$ is picked independently and with replacement) We use $D \sim \mathsf{LS}(A)$ to denote that.*

### 3.3 Statistical Dimension

In addition to leverage scores capturing the geometric influence of matrix rows, another related notion that will play an important role is the statistical dimension. While leverage scores are row-specific concepts, statistical dimension provides an aggregate measure of the complexity of a matrix that governs the sample size needed to effectively sketch it.

**Definition 3.8** (Statistical dimension, see Definition 1 in Avron et al. (2017) as an example)**.** *For $\lambda \geq 0$ and rank-$d$ matrix $A \in \mathbb{R}^{n \times d}$ with singular values $\sigma_i(A)$, the quantity $\mathsf{sd}_\lambda(A) := \sum_{i=1}^d \frac{1}{1 + \lambda/\sigma_i(A)^2}$ is the statistical dimension of the ridge regression problem with regularizing weight $\lambda$.*

### 3.4 QUANTUM TOOLS FOR SUBSPACE EMBEDDING

Now, we construct a sampling matrix from estimated scores that serve as a subspace embedding.

**Lemma 3.9** (Informal Version of Lemma A.2). *Consider query access to matrix $A \in \mathbb{R}^{n \times d}$ with row sparsity $r$. Let $U$ denote the orthonormal basis of $A$. For any $\epsilon \in (0, 1)$, there is a quantum algorithm that returns a diagonal matrix $D \in \mathbb{R}^{n \times n}$ satisfying $\|D\|_0 = O(\epsilon^{-2} d \log d)$, $\|DUx\|_2^2 = (1 \pm \epsilon)\|Ux\|_2^2$ (subspace embedding), and $D \sim \mathsf{LS}(A)$ (see Definition 3.7). This quantum algorithm makes $\widetilde{O}(\sqrt{nd}/\epsilon)$ row queries to $A$ and*

$$\widetilde{O}(r\sqrt{nd}/\epsilon + d^\omega)$$

*time, with the success probability $0.999$.*

*Sketch of our proof.* We use Lemma A.1 to estimate the leverage scores $\sigma_i$ of the input matrix $A$ to constant precision in $\widetilde{O}(\sqrt{nd})$ time. Classically, it is known that if we sample $O(\epsilon^{-2} d \log d)$ rows according to the leverage score distribution, the subsampled matrix $D$ acts as an $\epsilon$-subspace embedding for $A$. Quantumly, we can perform this sampling using the estimated leverage scores. Sampling $k = \|D\|_0 = O(\epsilon^{-2} d \log d)$ rows requires $O(\sqrt{nk})$ row queries to $A$. The total runtime follows from the estimation cost in Lemma A.1 plus the sampling cost, which is $\widetilde{O}(r\sqrt{nd}/\epsilon + d^\omega)$. □

## 4 MULTIPLE REGRESSION AND LINEAR REGRESSION

In Section 4.1, we show that the leverage score distribution may imply the subspace embedding and approximate matrix product. In Section 4.2, we show that by using the subspace embedding and approximate matrix product, we get multiple regression. In Section 4.3, we analyze the running time for each of the matrices $SA$, $SB$, $(SA)^\dagger$, and $(SA)^\dagger \cdot (SB)$. In Section 4.4, we combine the important properties of this section to form the formal version of our result for multiple regression, and based on that, we take $N = 1$ to form the formal version of our result for linear regression.

### 4.1 LS IMPLIES SE AND AMP

In this section, we present a tool from Song et al. (2019c) showing that if $S \sim \mathsf{LS}(A)$, a leverage score distribution, then $S$ is a subspace embedding (see Definition 3.1) and satisfies the definition of Frobenius norm approximate matrix product (see Definition 3.2). The purpose is to rigorously justify why sampling from leverage scores enables dimensionality reduction for regression problems. By showing that the sampled matrix retains the structure of the original matrix, we lay the groundwork to prove that solving regression on the smaller sampled matrix yields an approximate solution for the full regression problem. This then sets up the development in later sections showing how this sampling-based reduction leads to faster quantum algorithms.

**Lemma 4.1** (Corollary C.30 in Song et al. (2019c)). *Given $A \in \mathbb{R}^{n \times d}$, we let $U$ denote the orthonormal basis of $A$, $S \sim \mathsf{LS}(A)$ (see Definition 3.7), and $\|S\|_0 = m$. If $m = O(d \log d)$, then $S$ is a $\mathsf{SE}(1/2, 0.99, n, d)$ subspace embedding (see Definition 3.1) for $U$. If $m = O(d/\epsilon)$, then $S$ satisfies $\mathsf{FAMP}(\sqrt{\epsilon/d}, 0.99, n, d)$ (see Definition 3.2) for $U$.*

### 4.2 FROM SE AND AMP TO REGRESSION

In this section, we present another tool from Song et al. (2019c) showing that if $S$ is a subspace embedding (see Definition 3.1) and satisfies the definition of Frobenius norm approximate matrix product (see Definition 3.2), then the multiple regression is satisfied. Therefore, solving the sketched regression problem on the smaller matrix $SA$ yields solutions that generalize to approximate the full regression problem on $A$.

**Lemma 4.2** (Lemma C.31 in Song et al. (2019c)). *Let $A \in \mathbb{R}^{n \times d}$, $B \in \mathbb{R}^{n \times N}$, $S \sim \mathsf{LS}(A)$ (see Definition 3.7), $X^* = \arg\min_X \|AX - B\|_F^2$, $X' = \arg\min_X \|SAX - SB\|_F^2$, and $U$ denote an orthonormal basis for $A$. If $S$ is $\mathsf{SE}(1/2, 0.99, n, d)$ (see Definition 3.1) and $\mathsf{FAMP}(\sqrt{\epsilon/d}, 0.99, n, d)$ (see Definition 3.2) for $U$, then we have*

$$\|AX' - B\|_F^2 \le (1 + \epsilon)\|AX^* - B\|_F^2.$$

### 4.3 COMPUTING THE RUNNING TIME

In this section, we state and prove a lemma bounding the running time for using leverage score sampling (see Definition 3.7), then forming the sketched matrices $SA$ and $SB$, computing the pseudoinverse of $SA$, and multiplying this pseudoinverse by $SB$ to obtain the sketched solution. Each of these pieces is analyzed in terms of the input dimension $n$, $d$, and the accuracy parameter $\epsilon$. The purpose of this section is to complement the correctness guarantees from Section 4.1 and Section 4.2 by quantifying the computational efficiency of the sampling reduction process.

**Lemma 4.3** (Informal Version of Lemma B.3). *Let $A \in \mathbb{R}^{n \times d}$, $B \in \mathbb{R}^{n \times N}$, $\epsilon \in (0, 0.1)$, $\omega \approx 2.37$, $S$ denote a diagonal matrix that $\|S\|_0 = O(d \log d + d/\epsilon)$, and $X' = (SA)^\dagger SB$. Then, we have we can compute $X' \in \mathbb{R}^{d \times N}$ in*

$$\widetilde{O}(d^\omega/\epsilon + Nd^{\omega-1}/\epsilon)$$

*time*.

*Sketch of our proof.* The key steps to analyze the running time of computing the sketched solution $X' = (SA)^\dagger SB$ are as follows. First, compute the sketch $SA$ in $\widetilde{O}(d^2/\epsilon)$ time, where $S$ is an $n \times n$ diagonal matrix with $\widetilde{O}(d/\epsilon)$ non-zero entries. Second, compute the sketch $SB$ in $\widetilde{O}(Nd/\epsilon)$ time. Third, compute $(SA)^\dagger$ in $\widetilde{O}(d^\omega/\epsilon)$ time, where $SA$ is a $d \times \widetilde{O}(d/\epsilon)$ matrix. Finally, compute $(SA)^\dagger(SB)$ in $\widetilde{O}(Nd^{\omega-1}/\epsilon)$ time using fast matrix multiplication (Fact A.6), where $SB$ is a $\widetilde{O}(d/\epsilon) \times N$ matrix. Thus, the total time to compute $X' = (SA)^\dagger SB$ is $\widetilde{O}(d^\omega/\epsilon + Nd^{\omega-1}/\epsilon)$. $\square$

### 4.4 MAIN RESULT

At this point, we have developed all the theoretical concepts required to obtain faster quantum algorithms for regression problems based on a sampling reduction approach. Thus, in this section, we present our main results for the multiple regression and the linear regression. First, we incorporate the mathematical properties developed earlier to present our result for the multiple regression.

**Theorem 4.4** (Quantum algorithm for multiple regression, restatement of Theorem 1.5). *Let $\epsilon \in (0, 1)$. Let $\omega \approx 2.37$ denote the exponent of matrix multiplication. Given a matrix $A \in \mathbb{R}^{n \times d}$ with row sparsity $r$, where $r \leq d$, $B \in \mathbb{R}^{n \times N}$, there is a quantum algorithm that outputs $X \in \mathbb{R}^{d \times N}$ such that*

$$\|AX - B\|_F \leq (1 + \epsilon) \min_{X' \in \mathbb{R}^{d \times N}} \|AX' - B\|_F,$$

*which takes $\widetilde{O}(\sqrt{nd}/\epsilon)$ row queries to $A$ and $\widetilde{O}(\sqrt{n}d^{1.5}/\epsilon + d^\omega/\epsilon + Nd^{\omega-1}/\epsilon)$ time. The success probability is 0.999.*

*Proof.* Note that by combining Lemma 4.1 and Lemma 4.2, we can have $\|AX' - B\|_F^2 \leq (1 + \epsilon)\|AX^* - B\|_F^2$.

Lemma 3.9 and Lemma 4.3 give us the running time. $\square$

Then, we present our result for the linear regression. This is the multiple regression with $N = 1$.

**Theorem 4.5** (Quantum algorithm for linear regression, restatement of Theorem 1.4). *Let $\epsilon \in (0, 1)$. Let $\omega \approx 2.37$ denote the exponent of matrix multiplication. Given a matrix $A \in \mathbb{R}^{n \times d}$ and $b \in \mathbb{R}^n$, there is a quantum algorithm that outputs $x \in \mathbb{R}^d$ such that*

$$\|Ax - b\|_2 \leq (1 + \epsilon) \min_{x' \in \mathbb{R}^d} \|Ax' - b\|_2,$$

*which takes $\widetilde{O}(\sqrt{nd}/\epsilon)$ row queries to $A$ and $\widetilde{O}(\sqrt{n}d^{1.5}/\epsilon + d^\omega/\epsilon)$ time, where $r$ is the row of sparsity of matrix $A$ and $r \leq d$. The success probability is 0.999.*

*Proof.* Let $x := X \in \mathbb{R}^{d \times N}$ when $N = 1$. Let $b := B \in \mathbb{R}^{n \times N}$ when $N = 1$. Then, by Theorem 4.4, we have $\|Ax - b\|_2 \leq (1 + \epsilon) \min_{x' \in \mathbb{R}^d} \|Ax' - b\|_2$, which takes $\widetilde{O}(r\sqrt{nd}/\epsilon + d^\omega/\epsilon + d^{\omega-1}/\epsilon) = \widetilde{O}(\sqrt{n}d^{1.5}/\epsilon + d^\omega/\epsilon)$ time. $\square$

## 5 RIDGE REGRESSION

In Section 5.1, we present a property of the orthonormal basis for the ridge matrix. In Section 5.2, we introduce a sampling oracle related to $U_1$. In Section 5.3, we present the property of the leverage score distribution that for a matrix $S$ sampled from it, $S$ is a subspace embedding (Definition 3.1) and satisfy the definition of the spectral norm approximate matrix product (Definition 3.3). In Section 5.4, we present the guarantee of the sketched solution. In Section 5.5, we present our main result for the ridge regression.

### 5.1 PROPERTY OF ORTHONORMAL BASIS FOR RIDGE MATRIX

To develop our quantum ridge regression algorithm, we require an efficient way to reduce the ridge regression problem on the original matrix $A$ to a regularized regression problem on a much smaller sampled matrix. Therefore, in this section, we present the property of the orthonormal basis for the ridge matrix, and our proof is delayed to Section B.2. Compared to Lemma 12 in Avron et al. (2017), our analysis of the orthonormal basis is the part that strengthens this lemma. Lemma 12 in Avron et al. (2017), on the other hand, provides an explicit formula for the squared Frobenius norm of $U_1$ in terms of the statistical dimension of the ridge problem. It also bounds the spectral norm of $U_1$, which will be useful for ensuring subspace embedding properties (see Definition 3.1) when we subsample the ridge leverage scores of $U_1$.

**Claim 5.1** (A stronger version of Lemma 12 in Avron et al. (2017)). *Given matrix $A \in \mathbb{R}^{n \times d}$, we let $U \in \mathbb{R}^{n \times d}$ denote the orthonormal basis of $A$, $U_1 \in \mathbb{R}^{n \times d}$ comprise the first $n$ rows of orthonormal basis of $\begin{bmatrix} A \\ \sqrt{\lambda}I_d \end{bmatrix}$, and for each $i \in [d]$, $\sigma_i(A)$ denote the singular value of matrix $A$. Then we have $\|U\|_F^2 = d$, $\|U\| = 1$, $\|U_1\|_F^2 = \sum_{i=1}^d \frac{1}{1+\lambda/\sigma_i(A)^2} = \mathsf{sd}_\lambda(A)$, and $\|U_1\| = \frac{1}{\sqrt{1+\lambda/\sigma_1^2}}$.*

### 5.2 SAMPLING ORACLE RELATED TO $U_1$

In this section, we present a sampling oracle related to $U_1$. Specifically, given query access to an $n \times d$ matrix $A$, we show how to sample rows from $A$ according to the ridge leverage score distribution of $U_1$ in input sparsity time. This sampling oracle serves as a crucial subroutine in our quantum ridge regression algorithm, enabling us to reduce solving the ridge regression problem on $A$ to solving a sampled regular regression problem on a much smaller sampled matrix.

**Lemma 5.2** (Sampling oracle). *Consider query access to matrix $A \in \mathbb{R}^{n \times d}$ with row sparsity $r$. Let $U_1$ denote the comprise of first $n$ row of orthonormal basis of $\widetilde{A} = \begin{bmatrix} A \\ \sqrt{\lambda}I_d \end{bmatrix}$. For any $\epsilon \in (0, 1)$, there is a quantum algorithm that, returns a diagonal matrix $D \in \mathbb{R}^{n \times n}$, where $\|D\|_0$ denote the sparsity of $D$, such that $D \sim \mathsf{LS}(\widetilde{A}_{1:n})$ (see Definition 3.7), where $\widetilde{A}_{1:n}$ is the distribution with respect to $U_1$, and this quantum algorithm makes $\widetilde{O}(\sqrt{n \cdot \|D\|_0})$ row queries to $A$ and takes*

$$\widetilde{O}(r\sqrt{n\|D\|_0} + \mathrm{poly}(d, \|D\|_0))$$

*time, with a success probability of $0.999$.*

*Proof.* The proof is similar to Lemma 3.9. The major difference is, in Lemma 3.9, we estimate a distribution with $n$ scores, and take samples from it. Here, we estimate a distribution $n + d$ scores (with respect to $\widetilde{A} \in \mathbb{R}^{(n+d) \times d}$), but we only take samples from first $n$ scores (with respect to $U_1 \in \mathbb{R}^{n \times d}$). Since the summation of first $n$ scores is fixed and can be explicitly computed (see Claim 5.1 for computation of $\|U_1\|_F^2$). Thus, our sampling is correct. □

### 5.3 FROM LS TO SE AND SAMP

With the sampling oracle and the structural ridge regression properties established in the previous sections, we now have the key ingredients to show the reduction from ridge regression on $A$ to subsampled regular regression. In this section, we present a tool from Avron, Clarkson, and Woodruff Avron et al. (2017), which shows that if $S$ is sampled from the leverage score distribution, then it

is a subspace embedding and spectral norm approximate matrix product. The formal guarantees showing that the ridge leverage score sampler $S$ satisfies the desired properties with respect to $U_1$ are presented in the following Lemma. This lemma helps establish the validity of our approach to subsample the ridge regression problem and solve the smaller sketched problem. In later sections, we show how solving this sketched regression problem leads to fast quantum algorithms for ridge regression.

**Lemma 5.3** (Theorem 16 in Avron et al. (2017)). *Given matrix $A \in \mathbb{R}^{n \times d}$, we let $U_1 \in \mathbb{R}^{n \times d}$ comprise the first $n$ rows of the orthonormal basis of $\begin{bmatrix} A \\ \sqrt{\lambda}I_d \end{bmatrix}$, $S \sim \mathsf{LS}(A)$, and $\|S\|_0 = \widetilde{O}(\epsilon^{-1}\mathsf{sd}_\lambda(A))$.*

*Then we have that $S$ is $\mathsf{SE}(1/2, 0.99, n, d)$ and $\mathsf{SAMP}(\sqrt{\epsilon}, 0.99, n, d)$ for $U_1$*

### 5.4 GUARANTEE OF SKETCHED SOLUTION

What remains is to formally argue that solving the sketched regression problem on the sampled matrix $SU_1$ yields a good approximation to the original ridge regression problem on $A$. We fill this missing step by showing that the guarantees provided by subspace embedding and spectral norm approximate matrix product on $U_1$ imply that the ridge regression objective is well-preserved between the original problem and the sketched problem.

**Lemma 5.4** (Lemma 11 in Avron et al. (2017)). *Given matrix $A \in \mathbb{R}^{n \times d}$, we let $U_1 \in \mathbb{R}^{n \times d}$ comprise the first $n$ rows of orthonormal basis of $\begin{bmatrix} A \\ \sqrt{\lambda}I_d \end{bmatrix}$. Suppose $S$ is $\mathsf{SE}(1/2, 0.99, n, d)$ and $\mathsf{SAMP}(\sqrt{\epsilon}, 0.99, n, d)$ for $U_1$. Then, we have $\|Ax' - b\|_2^2 + \lambda \|x'\|_2^2 \le (1+\epsilon) \cdot (\|Ax^* - b\|_2^2 + \lambda \|x^*\|_2^2)$.*

### 5.5 MAIN RESULT

Now we present our main result for the ridge regression, bringing together the sampling oracle and the reduction arguments based on subspace embedding and approximate matrix multiplication.

**Theorem 5.5** (Quantum algorithm for ridge regression, restatement of Theorem 1.6). *Given a matrix $A \in \mathbb{R}^{n \times d}$ and $b \in \mathbb{R}^n$, we let $\mathsf{sd}_\lambda(A)$ denote the statistical dimension of matrix $A$ (see Definition 3.8), $\epsilon \in (0, 1)$, and $\lambda > 0$ denote a regularization parameter. There is a quantum algorithm that outputs $x \in \mathbb{R}^d$ such that*

$$\|Ax - b\|_2^2 + \lambda \|x\|_2^2 \le (1 + \epsilon) \min_{x' \in \mathbb{R}^d} (\|Ax' - b\|_2^2 + \lambda \|x'\|_2^2),$$

*which takes $\widetilde{O}(\sqrt{n \cdot \mathsf{sd}_\lambda(A)}/\epsilon)$ row queries to $A$ and $\widetilde{O}(\sqrt{n \cdot \mathsf{sd}_\lambda(A)}d/\epsilon + \mathrm{poly}(d, \mathsf{sd}_\lambda(A), 1/\epsilon))$ time, with 0.999 success probability.*

*Proof.* It follows from combining Lemma 3.9, 5.2, 5.3, 5.4, and Claim 5.1. $\square$

## 6 CONCLUSION

In this paper, we present quantum algorithms for linear regression, multiple regression, and ridge regression that achieve quadratic speedups in the data dimension $n$ compared to classical algorithms (Clarkson & Woodruff, 2013; Nelson & Nguyên, 2013), without dependence on data-related parameters. Specifically, the quantum linear regression algorithm runs in $\widetilde{O}(\sqrt{n}d^{1.5}/\epsilon + d^\omega/\epsilon)$ time, improving over the running time of the classical algorithm $O(nd) + \mathrm{poly}(d/\epsilon)$. The multiple regression and ridge regression algorithms achieve similar quadratic improvements. Our key contribution is developing these accelerated quantum algorithms while removing their previous dependence on matrix-specific parameters like the condition number. Without removing the condition number, the quantum algorithm can only speed up over the classical algorithm when encountering well-conditioned matrices. Our algorithms rely on a sampling-based dimensionality reduction leveraging properties of leverage scores and statistical dimension. By reducing the original regression problem to one on a smaller subsampled matrix, quantum routines can solve this smaller problem and generalize the solution to the full data.

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

## A PRELIMINARIES

In Section A.1, we incorporate the quantum tools to study the properties of subspace embedding. In Section A.2, we show that the leverage score sample preserves the approximate matrix product. In Section A.3, we present the running times of fast matrix multiplication.

**Notations.** First, we introduce the notations related to the sets. We define $\mathbb{Z}^+ := \{1, 2, 3, \dots\}$ to be the set containing all positive integers. Let $n, d \in \mathbb{Z}^+$. We define $[n] := \{1, 2, 3, \dots, n\}$. We use $\mathbb{R}$, $\mathbb{R}^n$, and $\mathbb{R}^{n \times d}$ to denote the set containing all real numbers, all $n$-dimensional vectors with real entries, and the $n \times d$ matrices with real entries.

Now, we introduce the notations related to vectors. Let $x \in \mathbb{R}^n$. For all $i \in [n]$, we let $x_i \in \mathbb{R}$ be the $i$-th entry of $x$. We define the $\ell_2$ norm of $x$, denoted as $\|x\|_2$, as $\sqrt{\sum_{i=1}^n x_i^2}$.

After that, we present the notations related to the matrices. Let $A \in \mathbb{R}^{n \times d}$. For all $i \in [n]$, $j \in [d]$, we define $A_{i,j} \in \mathbb{R}$ as the entry of $A$ at the $i$-th row and $j$-th column; we define $A_{i,*} \in \mathbb{R}^d$ as the $i$-th row of $A$; we define $A_{*,j} \in \mathbb{R}^n$ as the $j$-th column of $A$. Given a vector $y \in \mathbb{R}^d$ satisfying $\|y\|_2 = 1$, we define the spectral norm of $A$, denoted as $\|A\|$, to be $\max_{y \in \mathbb{R}^d} \|Ay\|_2$. We define the Frobenius norm of $A$ as $\|A\|_F := \sqrt{\sum_{i=1}^n \sum_{j=i}^d |A_{i,j}|^2}$. The $\ell_0$ norm of $A$, denoted as $\|A\|_0 \in \mathbb{R}$, is defined to be the number of nonzero entries in $A$. We use $I_d$ to represent the $d \times d$ identity matrix. We use $A^\top \in \mathbb{R}^{d \times n}$ to denote the transpose of the matrix $A$. $A^\dagger$ denote the pseudoinverse of $A$. Given two symmetric matrices $B, C \in \mathbb{R}^{n \times n}$, we use $B \preceq C$ to represent that the matrix $C - B$ is positive semidefinite (or PSD), namely for all $x \in \mathbb{R}^n$, we have $x^\top (C - B) x \geq 0$.

Finally, we define the notations related to functions. We use $\text{poly}(n)$ to represent a polynomial in $n$. Let $f, g : \mathbb{R} \to \mathbb{R}$ be two functions. We use $\widetilde{O}(f)$ to denote $f \cdot \text{poly}(\log f)$. We use $g(n) = O(f(n))$ to represent that there exist two positive real numbers $C$ and $x_0$ such that for all $n \geq x_0$, we have $|g(n)| \leq C \cdot f(n)$. $\arg\min_x f(x)$ denote the $x$ value such that $f(x)$ attains its minimum.

### A.1 QUANTUM TOOLS FOR SUBSPACE EMBEDDING

In this section, our purpose is to bridge the gap between classical theory and quantum techniques: we present a quantum tool we use for designing a fast algorithm based on leverage score sampling. This tool was recently developed by Apers and Gribling Apers & Gribling (2023). In particular, we state two Lemmas analyzing efficient quantum routines for leverage score estimation and sampling, respectively. The first allows approximating row leverage scores (see Definition 3.5) with queries to the input matrix scaling as the square root of the dimension. The second one constructs a sampling matrix from these estimated scores that serve as a subspace embedding (see Definition 3.1).

**Lemma A.1** (Theorem 3.2 in Apers & Gribling (2023)). *Consider query access to matrix $A \in \mathbb{R}^{n \times d}$ with row sparsity $r$. For any $\epsilon_0 \in (0, 1)$, there exists a quantum algorithm that provides query access to estimate $\widetilde{\sigma}_i$ for any $i \in [n]$ satisfying $\widetilde{\sigma}_i = (1 \pm \epsilon_0)\sigma(A)_i$, with the following guarantees:*

- *The algorithm makes $\widetilde{O}(\sqrt{nd}/\epsilon_0)$ row queries to $A$.*

- *It runs in $\widetilde{O}(r\sqrt{nd}/\epsilon_0 + d^\omega/\epsilon_0^2 + d^2/\epsilon_0^4)$ time.*

- *The success probability is at least 0.999*

- *The cost per estimate $\widetilde{\sigma}_i$ is one row query to $A$ and $\widetilde{O}(r/\epsilon_0^2)$ time*

**Lemma A.2** (Formal Version of Lemma 3.9). *Consider query access to matrix $A \in \mathbb{R}^{n \times d}$ with row sparsity $r$. Let $U$ denote the orthonormal basis of $A$. For any $\epsilon \in (0, 1)$, there is a quantum algorithm that, returns a diagonal matrix $D \in \mathbb{R}^{n \times n}$ such that*

- $\|D\|_0 = O(\epsilon^{-2} d \log d)$

- $\|DUx\|_2^2 = (1 \pm \epsilon)\|Ux\|_2^2$ *(subspace embedding)*

- $D \sim \mathsf{LS}(A)$ *(see Definition 3.7)*

- *It makes $\widetilde{O}(\sqrt{nd}/\epsilon)$ row queries to $A$.*

- *It takes $\widetilde{O}(r\sqrt{nd}/\epsilon + d^\omega)$ time.*

- *The success probability* 0.999

*Proof.* To do the leverage score sampling, we only need to set $\epsilon_0 = 0.1$ to be constant in Lemma A.1.

Using classical correctness, we know that if the sampling size is $O(\epsilon^{-2}d\log d)$, then we will get subspace embedding (see Definition 3.1).

Using quantum sampling lemma, we know that sampling $\|D\|_0$ rows from an $n$ rows of $A$ requires $\sqrt{n\|D\|_0}$ row queries to $A$.

Using Lemma A.1 it takes

$$\widetilde{O}(r\sqrt{n\|D\|_0} + d^\omega)$$

Thus, we complete the proof. $\qquad\square$

### A.2 LEVERAGE SCORE SAMPLE PRESERVES APPROXIMATE MATRIX PRODUCT

In this section, we show that the leverage score sample preserves the approximate matrix product. Specifically, we analyze the Lemma showing that sampling rows of a matrix $A$ proportionally to leverage scores (see Definition 3.5) generates a sketching matrix that approximates the product between $A$ and any other matrix $B$ with respect to the Frobenius norm.

**Lemma A.3** (Lemma C.29 in Song et al. (2019c)). *If the following conditions hold*

- *Let $A \in \mathbb{R}^{n \times d}$*

- *Let $B \in \mathbb{R}^{n \times N}$*

- *Let $\epsilon \in (0, 1)$*

- *Let $S \sim \mathsf{LS}(A)$ (see Definition 3.7)*

- *Let $\|S\|_0 = O(1/\epsilon^2)$*

*Then, for any fixed matrix B, we have*

- *$\|A^\top S^\top SB - A^\top B\|_F^2 \le \epsilon^2 \|A\|_F^2 \|B\|_F^2$*

- *The success probability is* 0.999

### A.3 FAST MATRIX MULTIPLICATION

In this section, we present the running time of the fast matrix multiplication. We define the variable $\mathcal{T}_{\mathrm{mat}}$ to represent the time cost of multiplying two matrices of designated dimensions. Further, we use $\omega$ to refer to the matrix multiplication exponent governing the asymptotic scaling of these runtimes Williams (2012); Le Gall (2014); Alman & Williams (2021); Duan et al. (2022); Gall (2023); Williams et al. (2023). We also introduce useful facts about manipulating the matrix multiplication time function. These rules will assist with analyzing the runtimes of operations like computing $(SA)^\dagger SB$ which occur inside our regression analysis (see Lemma 4.3).

**Definition A.4.** *Given two matrices $a \times b$ size and $b \times c$, we use the $\mathcal{T}_{\mathrm{mat}}(a, b, c)$ to denote the time of multiplying $a \times b$ matrix with another $b \times c$.*

We use $\omega$ to denote the number that $\mathcal{T}_{\mathrm{mat}}(n, n, n) = n^\omega$.

**Fact A.5.** *Given three positive integers, we have*

$$\mathcal{T}_{\mathrm{mat}}(a,b,c) = O(\mathcal{T}_{\mathrm{mat}}(a,c,b)) = O(\mathcal{T}_{\mathrm{mat}}(b,a,c)) = O(\mathcal{T}_{\mathrm{mat}}(b,c,a)) = O(\mathcal{T}_{\mathrm{mat}}(c,a,b)) = O(\mathcal{T}_{\mathrm{mat}}(c,b,a))$$

**Fact A.6.** *Given $a,b,c,d$ are positive integers. Then we have*

- **Part 1.**

$$\mathcal{T}_{\mathrm{mat}}(a,b,c) = O(d \cdot \mathcal{T}_{\mathrm{mat}}(a/d,b,c))$$

- **Part 2.**

$$\mathcal{T}_{\mathrm{mat}}(a,b,c) = O(d \cdot \mathcal{T}_{\mathrm{mat}}(a,b/d,c))$$

- **Part 3.**

$$\mathcal{T}_{\mathrm{mat}}(a,b,c) = O(d \cdot \mathcal{T}_{\mathrm{mat}}(a,b,c/d))$$

## B    MISSING PROOFS

In Section B.1, we present the proof of Claim 3.4. In Section B.2, we present the proof of Claim 5.1. In Section B.3, we state the formal version of Lemma 4.3 and present its proof.

### B.1    PROOF OF CLAIM 3.4

In this section, we restate and prove Claim 3.4.

**Claim B.1** (Restatement of Claim 3.4). *Let $A \in \mathbb{R}^{n \times d}$, $U$ denote the orthonormal basis of $A$, and $D$ denote a diagonal matrix such that $\|DAx\|_2^2 = (1 \pm \epsilon)\|Ax\|_2^2$ for all $x$.*

*Then, we have*

$$\|DUx\|_2^2 = (1 \pm \epsilon)\|Ux\|_2^2$$

*Proof.* Let $R \in \mathbb{R}^{d \times d}$ denote the QR factorization of $A$. Then we have

$$A = UR$$

From $\|DAx\|_2^2 = (1 \pm \epsilon)\|Ax\|_2^2$, we know that

$$\|DURx\|_2^2 = (1 \pm \epsilon)\|URx\|_2^2, \forall x$$

Since $R$ is full rank, then we can replace $Rx$ by $y$ to obtain

$$\|DUy\|_2^2 = (1 \pm \epsilon)\|Uy\|_2^2, \forall y.$$

$\square$

### B.2    PROOF OF CLAIM 5.1

In this section, we restate and prove Claim 5.1.

**Claim B.2** (Restatement of Claim 5.1). *Given matrix $A \in \mathbb{R}^{n \times d}$, we let $U \in \mathbb{R}^{n \times d}$ denote the orthonormal basis of $A$, $U_1 \in \mathbb{R}^{n \times d}$ comprise the first $n$ rows of orthonormal basis of $\begin{bmatrix} A \\ \sqrt{\lambda}I_d \end{bmatrix}$, and for each $i \in [d]$, $\sigma_i(A)$ denote the singular value of matrix $A$.*

*Then we have*

- **Part 1.** $\|U\|_F^2 = d$

- **Part 2.** $\|U\| = 1$

- **Part 3.** $\|U_1\|_F^2 = \sum_{i=1}^{d} \frac{1}{1 + \lambda/\sigma_i(A)^2} = \mathsf{sd}_\lambda(A)$

- **Part 4.** $\|U_1\| = \frac{1}{\sqrt{1+\lambda/\sigma_1^2}}$

*Proof.* For $\|U\|_F^2 = d$, it trivially follows from the definition of the orthonormal basis.

We consider the SVD[1] of

$$A = U\Sigma V^\top, \tag{1}$$

where $U \in \mathbb{R}^{n \times n}$, $\Sigma \in \mathbb{R}^{n \times d}$ and $V \in \mathbb{R}^{d \times d}$.

We define

$$D := (\Sigma^\top \Sigma + \lambda I_d)^{-1/2}. \tag{2}$$

We define

$$\widehat{A} := \begin{bmatrix} U\Sigma D \\ V\sqrt{\lambda}D \end{bmatrix} \tag{3}$$

Then we have

$$\widehat{A}^\top \widehat{A} = I_d$$

For any $x$, we define $y$

$$y := D^{-1}V^\top x \tag{4}$$

Then, we have

$$\begin{aligned}
\widehat{A}y &= \begin{bmatrix} U\Sigma D \\ V\sqrt{\lambda}D \end{bmatrix} D^{-1}V^\top x \\
&= \begin{bmatrix} U\Sigma DD^{-1}V^\top \\ V\sqrt{\lambda}DD^{-1}V^\top \end{bmatrix} x \\
&= \begin{bmatrix} U\Sigma V^\top \\ \sqrt{\lambda}VV^\top \end{bmatrix} x \\
&= \begin{bmatrix} A \\ \sqrt{\lambda}I_d \end{bmatrix} x,
\end{aligned}$$

where the first step follows from the definition of $\widehat{A}$ (see Eq. (3)) and $y$ (see Eq. (4)), the second step follows from simple algebra, the third step follows from the fact that $DD^{-1}$ is the identity matrix, and the last step follows from the SVD of $A$ (see Eq. (1)) and the fact that $V$ is orthogonal.

Finally, we can show

$$\begin{aligned}
\|U_1\|_F^2 &= \|U\Sigma D\|_F^2 \\
&= \|\Sigma D\|_F^2 \\
&= \sum_{i=1}^d \frac{1}{1 + \lambda/\sigma_i(A)^2},
\end{aligned}$$

where the first step follows from the Lemma statement, the second step follows from $U$ is an $n \times n$ orthonormal basis, and the last step follows from the definition of statistical dimension (see Definition 3.8). □

## B.3 RUNNING TIME ANALYSIS

In this section, we analyze the running time by presenting the formal version of Lemma 4.3.

**Lemma B.3** (Formal Version of Lemma 4.3). *Let $A \in \mathbb{R}^{n \times d}$, $B \in \mathbb{R}^{n \times N}$, $\epsilon \in (0, 0.1)$, $\omega \approx 2.37$, $S$ denote a diagonal matrix that $\|S\|_0 = O(d \log d + d/\epsilon)$, and $X' = (SA)^\dagger SB$.*

*Then, we have we can compute $X' \in \mathbb{R}^{d \times N}$ in $\widetilde{O}(d^\omega/\epsilon + Nd^{\omega-1}/\epsilon)$*

---

[1]Here we use a different shape of SVD, which is not as usual $\Sigma \in \mathbb{R}^{d \times d}$

*Proof.* The proof directly follows from computing the time of $SA$, $SB$, $(SA)^\dagger$, and $(SA)^\dagger \cdot (SB)$.

Before computing the running time, let us recall the definition

- $S$ is an $n \times n$ size diagonal entries which only has $\widetilde{O}(d/\epsilon)$ non-entries on diagonal

- $A$ has size $n \times d$

Here are the computation costs:

- Computing $\widetilde{O}(d/\epsilon) \times d$ size matrix $SA$ takes $\widetilde{O}(d^2/\epsilon)$

- Computing $\widetilde{O}(d/\epsilon) \times N$ size matrix $SB$ takes $\widetilde{O}(Nd/\epsilon)$ time

- Computing $d \times \widetilde{O}(d/\epsilon)$ size matrix $(SA)^\dagger$ takes $\widetilde{O}(d^\omega/\epsilon)$ time

- Computing $d \times N$ size matrix $(SA)^\dagger(SB)$ takes $\mathcal{T}_{\mathrm{mat}}(d, \widetilde{O}(d/\epsilon), N) = \widetilde{O}(Nd^{\omega-1}/\epsilon)$ time (due to Fact A.6).

Thus, we complete the proof. $\square$

## C   MORE RELATED WORK

Sketching can also be adapted to an iterative process to reduce the cost of iteration. This is the so-called *Iterate-and-sketch* approach and it has led to fast algorithms for many fundamental problems, such as linear programming (Cohen et al., 2021; Song & Yu, 2021; Jiang et al., 2021; Liu et al., 2023), empirical risk minimization (Lee et al., 2019; Qin et al., 2023c), dynamic kernel estimation (Qin et al., 2022b), projection maintenance (Song et al., 2023c) semi-definite programming (Gu & Song, 2022), John Ellipsoid computation (Song et al., 2022c), Frank-Wolfe algorithm (Xu et al., 2021; Song et al., 2022a), reinforcement learning (Shrivastava et al., 2023), rational database (Qin et al., 2022a), matrix sensing (Qin et al., 2023d), softmax-inspired regression (Deng et al., 2023a; Gao et al., 2023c; Li et al., 2023; Sinha et al., 2023), submodular maximization (Qin et al., 2023a), federated learning (Song et al., 2023b; Bian et al., 2023; Gao et al., 2023b), discrepancy problem (Deng et al., 2022; Song et al., 2022b), non-convex optimization (Song et al., 2021b;c; Alman et al., 2023; Zhang, 2022).

