# OpenReview forum: "Revisiting Quantum Algorithms for Linear Regressions: Quadratic Speedups without Data-Dependent Parameters"
_ICLR.cc/2025/Conference — Submitted to ICLR 2025_

### Official Review · Reviewer_izjd · 2024-10-23

**Soundness:** 2
**Presentation:** 1
**Contribution:** 1
**Rating:** 3
**Confidence:** 4

**Summary:**

Linear regression is a fundamental problem in the field of optimization and linear algebra. In recent years, many quantum algorithms for linear regression problems have been proposed. However, the time complexity in some of the algorithms has a dependence on the condition number of the input matrix of size $n\times d$. In this work, the authors addressed this issue by proposing a new quantum algorithm with time complexity $O(\sqrt{n} d^{1.5}/\varepsilon + \text{poly}(d/\varepsilon))$ for obtaining an $\varepsilon$-approximate solution, which has no dependence on the condition number. The core technique to achieve this speedup is by using the quantum leverage score sampling procedure proposed in [AG23].

**Strengths:**

The authors applied the quantum leverage score sampling algorithm proposed in [AG23] to the linear regression problem, and obtained quantum algorithms for linear, multiple and ridge regression problems. The time complexity of their quantum algorithms has a square root dependence on the number of rows for the input matrix, a quadratic speedup in this parameter compared with their classical counterparts, at the cost of increasing extra dependence on the number of columns and the inverse of the precision parameters.

**Weaknesses:**

The work appears to lack novelty, as the primary techniques employed to develop the algorithm involve applying quantum leverage score sampling, as proposed in [AG23], and the classical leverage score sampling method for linear regression problems, as proposed in [CW13]. It appears that the paper directly replaces the leverage score sampling procedure from [CW13] with its quantum counterpart to achieve results. Furthermore, the concept of using a quantized leverage score sampling procedure for solving linear regression problems has previously been introduced in [Sha23].
Also, the result of the work seems a bit weak compared with the classical counterparts. For input an $n\times d$ matrix, a classical algorithm in [CW13] can run in time $O(nd \log (1/\varepsilon)) + \tilde{O} (d^3)$, which could have logarithmic dependence in the error parameter $\varepsilon$, but the quantum algorithm has a polynomial dependence on $\varepsilon$. For comparison, the quantum algorithm for linear programming proposed in [AG23] also achieves a speedup over their classical counterparts while keeping logarithmic dependence in the precision parameter $\varepsilon$.

**Questions:**

Same as the weaknesses part.

---

> ### Author Response · Authors · 2024-11-18
> **Thank you very much for your insightful comments!**
>
> We express our deepest gratitude to the reviewer for taking the time and effort to review our work. We thank you for pointing out the strengths of our work that we have attained a quadratic speed up in the parameters compared with the classical counterparts. Below, we want to address your concerns about our work.
>
> Regarding our technical contribution, while we indeed build upon quantum leverage score sampling [AG23] and classical leverage score sampling [CW13], our contribution goes significantly beyond simply replacing classical sampling with quantum sampling. We would like to highlight that our work discovers an interesting application of the recently developed quantum spectral approximation procedure. More importantly, we find that using this procedure to solve the linear regression problem can achieve state-of-the-art complexity in the tall, dense regime, and can overcome the condition number issue in almost all previous quantum linear regression solvers. This removal of the matrix condition number $\kappa (A)$ is very crucial as it was a major limitation of previous quantum algorithms. This is a significant technical achievement that requires careful analysis of the interaction between quantum sampling and regression problems. Prior quantum algorithms could only guarantee speedups for well-conditioned matrices. Our algorithm provides the first unconditional quantum speedup for these regression problems.
>
> Compared with [Sha23], we note that [Sha23] still has runtime dependencies on matrix-specific parameters. Their algorithm relies on the "cost of encoding a matrix into a quantum computer". Only if this cost is small, then their quantum algorithm achieves a quadratic speedup over classical algorithms. However, we achieve a quadratic speedup over classical algorithms without such dependencies. We provide comprehensive analysis for multiple regression and ridge regression variants, which are not addressed in [Sha23]. Our techniques yield better runtime bounds even for basic linear regression.
>
>
> Regarding the inverse polynomial dependence on $\epsilon$, it is quite common in classical sketching algorithms. Intuitively, to construct an $\epsilon$-subspace embedding (or even just applying the JL lemma), the number of rows of the sketched matrix must be at least $1/\epsilon^2$. Therefore, if we want to keep a high-accuracy spectral approximation, the complexity will depend on $\mathrm{poly}(1/\epsilon)$. However, for the linear regression problem, there exists some classical algorithms that can achieve $\log(1/\epsilon)$. For example, [CW13] show that leverage score sketching can be combined with some iterative methods. More specifically, they use an $O(1)$-accuracy sketching matrix to construct a preconditioner. Then, they prove that $O(\log(1/\epsilon))$ iterations of the Conjugate Gradient method suffice to produce an $\epsilon$-error solution. The total time complexity is $\tilde{O}(\mathrm{nnz}(A)+d^3)$. We note that our quantum algorithm cannot be directly generalized to this setting. It can efficiently produce the preconditioner. However, the Conjugate Gradient method is hard to be quantumly sped up. We note that all the previous quantum linear regression algorithms have a $\mathrm{poly}(1/\epsilon)$ factor if a classical solution is required. Specifically, they are based on the quantum linear system solver, which outputs a quantum state that encodes the normalized solution. To obtain the classical solution, we need to perform a full state tomography and also need to estimate the normalization factor. These two steps must incur a $\mathrm{poly}(1/\epsilon)$ factor. We think achieving $\log(1/\epsilon)$ for the quantum linear regression algorithm is a very interesting open question.
>
> We sincerely thank you again for your insightful comments, and we have responded to all the points you have made. We hope that our response may address all of your concerns about our paper. Also, we have uploaded the updated version of our paper to OpenReview. We have marked our changes in red color.

---

> > ### Comment · Reviewer_izjd · 2024-11-26
> >
> > Thank you for your detailed rebuttal. While I sincerely appreciate your thorough response and carefully considered arguments, after thoughtful re-examination, I must maintain my original assessment regarding the lack of novelty. Therefore, I cannot support the acceptance of this submission.

---

### Official Review · Reviewer_Tkz2 · 2024-11-03

**Soundness:** 4
**Presentation:** 3
**Contribution:** 4
**Rating:** 8
**Confidence:** 4

**Summary:**

The paper presents a quantum algorithm for linear regression, with the running time of O(\sqrt{n}d^{1.5}+d^w) where n is the number of data points, d is the number of variables and w is the matrix multiplication exponent. It also gives quantum algorithms for multiple regression and ridge regression based on the same ideas.

Classically, running time must be at least linear in n. Quantumly, algorithms with running time polylogarithmic in n are known but their running time depends on the condition number of the regression matrix and other extra parameters. The quantum algorithm in the current paper is the first one with a running time that is sublinear in n, polynomial in d and 1/\epsilon where \epsilon is the desired accuracy and no dependence on extra parameters.

On the technical level, the results are based on combining the recent quantum algorithm by Apers and Gribling for computing leverage scores with classical algorithms for linear regression.

**Strengths:**

Natural research question: linear regression is an important computational task.
Clear improvement over previous work in the chosen parameter (n).
Writing generally good.

**Weaknesses:**

The paper is a combination of existing results from quantum algorithms (Apers-Gribling) and classical algorithms for linear regression but, given that the end result is interesting, I do not view this as a significant weakness.

**Questions:**

- Theorems 1.4 and 1.5 define that r is the row-sparsity of the matrix but the running time of the algorithms in them has no dependence on r. Consider removing r from them.
- "Classical linear algebra" section in the related work mentions a lot of references. It would be sufficient to give a smaller number of applications for classical linear algebra, as its significance is well understood.

---

> ### Author Response · Authors · 2024-11-18
> **Thank you very much for your insightful comments!**
>
> We express our deepest gratitude to the reviewer for taking the time and effort to review our work. We thank you for pointing out that our work studies an important question. It is very encouraging to see that our work has very clear improvement over the previous ones and our writing is generally good.
>
> Regarding the novelty, we would like to highlight that our work discovers an interesting application of the recently developed quantum spectral approximation procedure. More importantly, we find that using this procedure to solve the linear regression problem can achieve state-of-the-art complexity in the tall, dense regime, and can overcome the condition number issue in almost all previous quantum linear regression solvers. Moreover, our quantum algorithm’s time complexity matches the quantum lower bound in terms of $n$. Since linear regression is one of the most fundamental problems in optimization and machine learning, our work sets a new benchmark for it. We provide a tool to solve this problem with currently the best efficiency and fully classical outputs, which will be very helpful to use as a black box to solve other problems in the future.
>
> Question: Theorems 1.4 and 1.5 define that $r$ is the row-sparsity of the matrix but the running time of the algorithms in them has no dependence on $r$. Consider removing $r$ from them.
>
> Answer: Thank you for pointing this out. The row sparsity $r$ does play a role in the running time. For example, the total running time of Theorem 1.5 should be $\widetilde{O}(r \sqrt{nd} / \epsilon + d^{\omega} / \epsilon + N d^{\omega -1} / \epsilon )$ (combining Lemma 3.9 and Lemma 4.3). Since the row sparsity $r$ must be smaller than or equal to the number of columns $d$, in the running time, we replace $r$ by $d$ to make it more clear. Therefore, instead of removing $r$, we added the condition $r \leq d$ to the theorem statement to make it more clear.
>
> Question: "Classical linear algebra" section in the related work mentions a lot of references. It would be sufficient to give a smaller number of applications for classical linear algebra, as its significance is well understood.
>
> Answer: We completely agree with this and have moved the related work of the iterate-and-sketch to the appendix. In the main text, we only keep the sketch-and-solve which is directly related to our work.
>
> We sincerely thank you again for your insightful comments, and we have responded to all the points you have made. We hope that our response may address all of your concerns about our paper. Also, we have uploaded the updated version of our paper to OpenReview. We have marked our changes in red color.

---

### Official Review · Reviewer_Yhid · 2024-11-04

**Soundness:** 4
**Presentation:** 3
**Contribution:** 2
**Rating:** 3
**Confidence:** 4

**Summary:**

This paper considers the problem of solving linear regression, multiple regression, and ridge linear regression using quantum algorithms. Previous quantum algorithms that leverage quantum linear algebra techniques have shown exponential speedups in the system size $n$, but they often depend on certain quantum linear algebra-related parameters, such as the condition number of the regression matrix. An open question is whether quantum speedups can be achieved without relying on data-dependent parameters. This paper answers this question in the affirmative by developing quantum algorithms for linear regression, multiple regression, and ridge regression with quadratic quantum speedups. Technically, the algorithms first generate a subspace embedding of the original regression matrix through sampling from its leverage score distribution, achievable in $O(\sqrt{n})$ time if given a quantum row query oracle. The algorithms then solve the regression problems within this reduced subspace, where the solutions also have small regrets in the original problems.

**Strengths:**

This paper develops algorithms that achieve quantum speedups without relying on data-dependent parameters, addressing a significant open problem in quantum linear systems research. The literature review is thorough and well-organized.

**Weaknesses:**

In my view, the techniques used in this paper are relatively straightforward. Specifically, the main result, Theorem 4.4 in Section 4, is proved by combining Lemma A.1, Lemma 4.1, Lemma 4.2, and Lemma 4.3. Among these, Lemmas 4.1 and 4.2 are from existing literature, Lemma 4.3 involves algebraic calculations on the runtime of matrix multiplication.

Minor comments:
1. There appears to be a grammatical issue in Lemma A.1.
2. Lemma 4.2 is missing the phrase "time" or "time complexity."

**Questions:**

I don't have further questions other than the existing ones above.

---

> ### Author Response · Authors · 2024-11-18
> **Thank you very much for your insightful comments!**
>
> We express our deepest gratitude to the reviewer for taking the time and effort to review our work. We thank you for pointing out that our work is well-organized and addresses a significant open problem in quantum linear systems research. Below, we want to address your concerns.
>
> While individual lemmas come from prior work, their combination to achieve quantum speedups without matrix-dependent parameters required significant technical innovation. The key challenge was developing a quantum-compatible sampling framework that preserves both subspace embedding and approximate matrix product properties while being efficiently implementable on quantum hardware. This required careful analysis to ensure the quantum estimation of leverage scores (Lemma A.1) could interface correctly with the classical sampling guarantees (Lemmas 4.1 and 4.2).
>
> Our framework provides the first quantum speedups for regression problems that work for any input matrix, not just well-conditioned ones. This required careful technical work to remove condition number dependencies while maintaining quantum advantages. On the one hand, it discovers an interesting application of the recently developed quantum spectral approximation algorithm. On the other hand, it sets a new benchmark for quantum optimization and machine learning algorithms since linear regression is a fundamental problem, and our algorithm overcomes the condition number issue in almost all previous quantum linear regression algorithms. Moreover, it provides a useful technical tool that can be easily applied as a black box for other applications.
>
> Regarding the minor comments, we have fixed the grammatical issue in Lemma A.1. Also, we have uploaded the updated version of our paper to OpenReview. We have marked our changes in red color.
>
> Regarding Lemma 4.2 (Lemma C.31 in Song et al. (2019c)), this is a correctness guarantee rather than an algorithmic result, so it doesn't have an associated time complexity.
>
> We sincerely thank you again for your insightful comments, and we have responded to all the points you have made. We hope that our response may address all of your concerns about our paper.

---

> > ### Comment · Reviewer_Yhid · 2024-11-27
> >
> > Thank you very much for your thorough and thoughtful rebuttal. While I greatly appreciate the effort and clarity in your response, I remain concerned about the novelty of the work and, therefore, respectfully maintain my rating.

---

### Meta-Review · Area_Chair_eVCp · 2024-12-17

**Metareview:**

This paper considers solving the classical problem of linear regression, multiple regression, and ridge linear regression using quantum algorithms. Previous quantum algorithms that leverage quantum linear algebra techniques have shown exponential speedups in the system size $n$, but they often depend on certain quantum linear algebra-related parameters, such as the condition number of the regression matrix. This paper seeks to circumvent this dependence.

However, two reviewers acknowledge that the technical contributions and novelty here are limited. They apply quantum leverage score sampling, as proposed in [AG23], and the classical leverage score sampling method for linear regression problems, as proposed in [CW13]. The authors combine these in a rather straightforward manner, and even after the discussion period, the reviewers remain unconvinced by the overall novelty of the paper. Hence, I have to unfortunately, reject the paper.

**Additional Comments On Reviewer Discussion:**

There were some discussions between the authors and reviewers during the discussion period. The authors tried to justify that even though some ingredients of their arguments were present in existing literature, putting them together to remove the dependence on the condition number is non-trivial. The AC is personally not an expert in the area (quantum algorithms), and hence, cannot judge the level of novelty. Hence, I have to defer to the two reviewers (who gave scores of 3) and who maintained that the overall novelty of the paper does not meet the standards for publication at ICLR.

---

### Decision · Program_Chairs · 2025-01-22

Reject